# Phosphatidic Acid Accumulates at Areas of Curvature in Tubulated Lipid Bilayers and Liposomes

**DOI:** 10.3390/biom12111707

**Published:** 2022-11-17

**Authors:** Broderick L. Bills, Michelle K. Knowles

**Affiliations:** 1Department of Chemistry and Biochemistry, University of Denver, Denver, CO 80210, USA; 2Molecular and Cellular Biophysics Program, University of Denver, Denver, CO 80210, USA

**Keywords:** phosphatidic acid, membrane curvature, tubules

## Abstract

Phosphatidic acid (PA) is a signaling lipid that is produced enzymatically from phosphatidylcholine (PC), lysophosphatidic acid, or diacylglycerol. Compared to PC, PA lacks a choline moiety on the headgroup, making the headgroup smaller than that of PC and PA, and PA has a net negative charge. Unlike the cylindrical geometry of PC, PA, with its small headgroup relative to the two fatty acid tails, is proposed to support negatively curved membranes. Thus, PA is thought to play a role in a variety of biological processes that involve bending membranes, such as the formation of intraluminal vesicles in multivesicular bodies and membrane fusion. Using supported tubulated lipid bilayers (STuBs), the extent to which PA localizes to curved membranes was determined. STuBs were created via liposome deposition with varying concentrations of NaCl (500 mM to 1 M) on glass to form supported bilayers with connected tubules. The location of fluorescently labeled lipids relative to tubules was determined by imaging with total internal reflection or confocal fluorescence microscopy. The accumulation of various forms of PA (with acyl chains of 16:0-6:0, 16:0-12:0, 18:1-12:0) were compared to PC and the headgroup labeled phosphatidylethanolamine (PE), a lipid that has been shown to accumulate at regions of curvature. PA and PE accumulated more at tubules and led to the formation of more tubules than PC. Using large unilamellar liposomes in a dye-quenching assay, the location of the headgroup labeled PE was determined to be mostly on the outer, positively curved leaflet, whereas the tail labeled PA was located more on the inner, negatively curved leaflet. This study demonstrates that PA localizes to regions of negative curvature in liposomes and supports the formation of curved, tubulated membranes. This is one way that PA could be involved with curvature formation during a variety of cell processes.

## 1. Introduction

Phosphatidic acid (PA) is a lipid that can be produced from phosphatidylcholine (PC), diacylglycerol, lysophosphatidic acid (LPA), or from glycerol 3-phosphate or dihydroxyacetone phosphate de novo [1,2]. The roles of PA are currently best described in plants, particularly *Arabidopsis thaliana*, where PA is involved in a wide variety of processes, from stress response to growth, acting primarily as a signaling molecule [3]. In yeast, PA is involved in sporulation and secretion [3]. In mammals, PA is involved in exocytosis, intraluminal formation, cell proliferation, signaling, tumor progression, and cell differentiation [3].

Many of the roles that PA have in biology are hypothesized to be related to the geometry of PA itself. PA has been shown to regulate cellular functions by altering membrane shape locally on the plasma membrane or organelle membranes [4]. PA has an inverse conical shape when compared to PC [5]. Inverse-conical-shaped lipids are predicted to ease membrane bending by fitting well into negatively curved membranes, whereas conical-shaped lipids, such as LPA [5] and PE with a large fluorescent dye label on the headgroup, are predicted to sort into positively curved membrane areas [6]. The presence of lipids with an intrinsic membrane shape makes membrane bending easier and could also be a mechanism by which lipids are sorted [7,8].

Several studies have focused on the interplay between membrane shape and intrinsic lipid shape. Positively curved membranes accumulate single-tailed lipids, such as fluorescein-labeled hexadecanoic acid [6] and LPA [9], which are conical in shape. Interestingly, lipids with two tails also accumulate (fluorescein 1,2-dihexadecanoyl-sn-glycero-3-phosphoethanolamine, Fl-DHPE) to a higher extent [6,10]. An alternative model for lipid sorting depends on the defect sites in the bilayer that are present when a membrane bends [11]. The positively curved side has packing defect sites that, when present, allow for the insertion of hydrophobic protein motifs or lipids [10,12]. Therefore, defect sites are places in the membrane that can be stabilized by being filled with lipids. Lipids with more carbons in the tails (either longer tails or two tails) have been shown to accumulate more at these sites [6,10]. This is in contrast to the concept of sorting based on the intrinsic lipid shape, and the work presented here focuses on both the role of lipid tails and the small headgroup in PA curvature membrane sorting.

To measure how lipids and proteins are recruited by or induce curved membranes, several model curved membrane assays exist, such as curved supported lipid bilayers [13], tubules [14], membrane-coated nanopatterned surfaces [15,16,17], and small liposome-based assays [10,18]. Recently, an assay using NaCl to induce tubule formation in supported lipid bilayers (SLBs) was developed by Schenk et al. [19]. These supported tubulated bilayers (STuBs) were used to study Sar1B, a vesicle-budding protein [19], but in this study, we demonstrate that this assay is also useful for measuring lipid accumulation at curvature. This assay was chosen partly due to the simplicity of creating tubules but also because the tubes are connected to the flat supported bilayer, allowing the measurement of recruitment of the curved regions to be directly compared to the flat areas in the same measurement.

In this work, STuBs were used to measure the recruitment of PA to curvature regions, to determine if PA supports membrane curvature formation, and to determine if PA accumulates at regions of curvature. STuBs were created from POPC, DOPE-PEG, and a fluorescent lipid (either PA, PC, or PE) and imaged using confocal and TIRF microscopies. From the image data, the number of tubes per area and the intensity of dye-labeled lipids per tube were quantified relative to the flat areas of the same samples. As an alternative, extruded, large unilamellar liposomes were measured in bulk and the accessibility of the dye to quenching agents was assessed to determine which leaflet accumulated PA. Our results demonstrate that PA stabilizes curvature and prefers negatively curved areas, relative to PC controls.

## 2. Materials and Methods

### 2.1. Liposome and STuBs Assembly

Lipids were mixed in chloroform (Pharmco-Aaper, Brookfield, CT, USA) to specified concentrations using glass syringes for a total of 250 nmol. Chloroform was evaporated using nitrogen and vacuum. Lipids were resuspended in 2 mL of buffer containing 140 mM KCl, 20 mM HEPES, and varying concentrations of NaCl (ThermoFisher, Waltham, MA, USA) at pH 7.4. The solution was probe sonicated for 5 min on ice. For SLBs, 8 well dishes (Cellvis, Mountain View, CA, USA) were cleaned by submersion in 0.1% SDS (ThermoFisher, Waltham, MA, USA) for 1 h, followed by 1% bleach overnight. Then, 100 µL of 2% Hellmanex was added to each well for one hour. Afterwards, the wells were rinsed three times in a buffer containing 140 mM KCl, 20 mM HEPES, and varying concentrations of NaCl at pH 7.4. A total of 100 µL of liposome stocks were deposited per well in 8 well dishes and incubated at 37°C for 1 h. SLBs were then imaged immediately. The lipids used in this study include 1-palmitoyl-2-oleoyl-glycero-3-phosphocholine (POPC, Avanti Polar Lipids, Alabaster, AL, USA, part #: 850457), 1,2-dioleoyl-sn-glycero-3-phosphoethanolamine-N-[methoxy(polyethylene glycol)-2000] (DOPE-PEG, Avanti Polar Lipids, Alabaster, AL, USA, part #: 880234), Marina Blue- 1,2-dihexadecanoyl-sn-glycero-3-phosphoethanolamine (MB-DPPE, ThermoFisher, Waltham, MA, USA, part #: M12652), 1,1′-dioctadecyl-3,3,3′,3′-tetramethylindodicarbocyanine, 4-chlorobenzenesulfonate salt (DiD, ThermoFisher, Waltham, MA, USA, part #: D7757), 1,1′-dioctadecyl-3,3,3′,3′-tetramethylindocarbocyanine perchlorate (DiI, ThermoFisher, Waltham, MA, USA, part #: D282), 1-palmitoyl-2-{6-[(7-nitro-2-1,3-benzoxadiazol-4-yl)amino]hexanoyl}-sn-glycero-3-phosphate (16:0 6:0-NBD PA, Avanti Polar Lipids, Alabaster, AL, USA, part #: 810173), 1-palmitoyl-2-{12-[(7-nitro-2-1,3-benzoxadiazol-4-yl)amino]dodecanoyl}-sn-glycero-3-phosphate (16:0 12:0-NBD PA Avanti Polar Lipids, Alabaster, AL, USA, part #: 810174), 1-oleoyl-2-{12-[(7-nitro-2-1,3-benzoxadiazol-4-yl)amino]dodecanoyl}-sn-glycero-3-phosphate (18:1 12:0-NBD PA, Avanti Polar Lipids, Alabaster, AL, USA, part #: 810176), 1-oleoyl-2-[12-[(7-nitro-2-1,3-benzoxadiazol-4-yl)amino]dodecanoyl]-sn-glycero-3-phosphocholine (18:1 12:0-NBD PC, Avanti Polar Lipids, Alabaster, AL, USA, part #: 810133), and 1,2-dipalmitoyl-sn-glycero-3-phosphoethanolamine-N-(7-nitro-2-1,3-benzoxadiazol-4-yl) (DPPE-NBD, Avanti Polar Lipids, Alabaster, AL, USA, part #: 810144). The liposomes contained 98% POPC, 1% DOPE-PEG, and 1% NBD-labeled lipids. The SLBs used for FRAP contained 96.9% POPC 1% DOPE-PEG, 2% MB-DPPE, and 0.1% DiD. The SLBs used for TIRF microscopy contained 98% POPC, 1% DOPE-PEG, and 1% NBD-labeled lipid. MB-DPPE concentration was 2%, the same as in past work [16], and NBD-labeled PA lipids were used at a concentration relevant to what is observed in cells [20,21]. POPC concentration was adjusted to accommodate the difference. All reagents were purchased from Sigma Aldrich (St. Louis, MO, USA), unless otherwise specified. DiI was used in TIRF measurements and DiD was used for confocal measurements of the membrane tubule size due to the excitation wavelengths available on different microscopes. MB-DPPE was used to measure fluidity.

### 2.2. Fluorescence Recovery after Photobleaching (FRAP)

FRAP was performed on a point-scanning confocal microscope (Olympus Fluoview 3000) to test the fluidity of the bilayers and the connectivity of the tubules. The FRAP region was a 60-pixel diameter (6.01 µm) circle. A 640 nm laser was used for DiD and a 405 nm laser was used for MB-DPPE. The imaging rate was 2.17 s per frame during 2-color imaging. A total of 3 frames were taken prior to bleaching, then the FRAP region was bleached for 1s, followed by 45 frames for recovery. FRAP occurred at room temperature, 20–22°C. FRAP data was corrected for photobleaching and normalized to the average of the pre-bleach frames as described in previous work [6,16]. Graphpad Prism v9.4.1 (San Diego, CA, USA) was used for plotting, fitting, and *t*-testing.

### 2.3. Total Internal Reflection Fluorescence Microscopy

STuBs samples were imaged immediately after assembly. Two-color imaging data were taken with a 60x (1.49 NA) objective followed by a 2.5x magnifying lens to obtain a magnification that was 0.109 µm/pixel on the detector (EMCCD, Andor iXon897, Abingdon, UK). A DualView (Optical Insights, Suwanee, GA, USA) was used to split the red and green fluorescence (565LP dichroic with 525/50 and 605/75 emission filters, Chroma Technologies, Bellows Falls, VT, USA) into separate channels onto the camera. Movies were taken at 1 frame/second using Micromanager v1.4 [22].

For image analyses, the red and green images were aligned using 200 nm carboxylate modified, yellow-green fluospheres (ThermoFisher, Waltham, MA, USA, part #: F8811) and a home-built alignment code in MATLAB (vR2021b, Natick, MA, USA), as used previously [23,24]. These nanoparticles (diameter = 200 nm) were also used to identify the diffraction limit of the TIRF in full-width half-max (FWHM) calculations [21,22]. Tubule positions were located by bandpass filtering with 9 pixels, followed by spot finding with a pixel size of 5 and a variable threshold. All spot-finding code was initially written in IDL [25], then made available in MATLAB [26]. Radial plots were calculated as described previously [27], and the intensity from each peak was normalized to an intensity of ~500nm away. These plots were used to calculate the FWHM of the tubules based on the max intensity and the flat intensity, also ~500nm away. The FWHM was called the diameter of the tubule for all plots. Intensity (ΔF/S) measurements were calculated, where ΔF = circle–annulus. The average intensity of a circle with a 5-pixel diameter and the intensity of an annulus 1-pixel wide, at 7 pixels from the peak, were used (Appendix A). This was normalized by S where S = annulus–background; the background was defined as the intensity of a membrane with no fluorophores present. This was described in detail previously [23]. Longer-length tubules were occasionally noted as others observed previously [19]. These tubules are typically disconnected from the bilayer and are rare. Therefore, these were not used in the measurements for this paper. All analyses were performed in MATLAB and code will be made available upon request. All statistical testing was performed in GraphPad Prism.

### 2.4. Fluorimetry

NBD-labeled liposomes in a 2.5 mL buffer (140 mM KCl, 20 mM HEPES, and 15mM NaCl at pH 7.4) were extruded through 100 nm filters and put in a 1 cm quartz cuvette with a stir bar. A total of 250 µL of 0.1 mM dithionite was added after 30 s. Readings were taken on a Cary Eclipse (Agilent, Santa Clara, CA, USA) once per second, with 0.1 s exposure, an excitation wavelength of 463 nm, and an emission of 533 nm. Excitation and emission slit widths were variable depending on the intensity of the samples, between 5 and 20 nm.

## 3. Results

### 3.1. The Formation of STuBs Depends on NaCl Concentration

To determine what concentration of NaCl reliably forms tubules, SLBs were formed using 96.9% POPC, 1% DOPE-PEG, 2% Marina Blue-DPPE, and 0.1% DiD at varying concentrations of NaCl (Figure 1). The lowest and highest concentrations were chosen to replicate the original STuBs study [19]. This combination of lipids was intended to be a simple mixture that mimics the plasma membrane and includes PEGylated lipids to cushion the bilayer from the glass surface [28] and mimic the crowded cellular environment [29]. Increasing the concentration of NaCl reproducibly induced the formation of more tubules (Figure 1A,B). On rare occasions, long tubules form as seen in other labs [19], but these were not included in our analyses due to their rarity and because their shapes complicate analysis. Instead, smaller tubules were measured which were fluid with the rest of the bilayer (Appendix A), suggesting that they are connected structures, rather than liposomes. These tubule structures recovered to the same extent as the flat regions of the membrane (Appendix A). The tubules were diffraction-limited (Figure 1A,E), but the intensity of the tubules varied with NaCl concentration (Figure 1D) based on measurements of ΔF/S, as described in Appendix A. Specifically, tubules assembled in 1000 mM NaCl were significantly more intense than those at 500 or 750 mM (Figure 1D), while differences in the FWHM values of the tubules at any [NaCl] were not significant (Figure 1E). This suggests that tubules may be larger with higher concentrations of NaCl. Because single tubules can be observed in this assay, the distribution of intensity and sizes can also be quantified (Figure 1G). The distributions of intensity (ΔF/S) and tubule size (FWHM) is similar for tubules prepared with 500, 750, or 1000 mM NaCl.

### 3.2. PA and DPPE Form More Tubules Than PC

To determine if lipids with intrinsic curvature affect the formation of STuBs or accumulate in certain regions, a tail labeled PA (NBD-PA) and a headgroup labeled DPPE (DPPE-NBD) were incorporated separately into the STuBs. In this experiment, all lipids were identical, except the ones labeled on the axis in Figure 2. Membranes were formed in a 1000 mM NaCl buffer, using 98% POPC, 1% DOPE-PEG, and 1% either DPPE-NBD, (18:1 12:0-NBD)-PC, or (18:1 12:0-NBD)-PA, then imaged using TIRF microscopy. Example images are shown alongside an average of the tubule regions for each lipid combination (Figure 2A). In this experiment, PC acted as a negative control, while DPPE-NBD was a positive control because it sorts into positively curved membranes [6,30]. The NBD-PA produced more tubules than NBD-PC and as many as DPPE-NBD (Figure 2B). This supports the hypothesis that PA affects membrane curvature formation. Next, the intensity of the tubules was evaluated to determine if the tubules contained higher amounts of the NBD-labeled lipids. The intensity (ΔF/S) varied as a function of the lipid headgroups with NBD-PA tubules being dimmer than the NBD-PC tubules (Figure 2C). The higher intensity (Figure 2C) either suggests that more PC is present in the tubules, relative to PA and DPPE, or that the tubules are larger when NBD-PC is present. However, no significant difference in the tubule diameter (FWHM) was observed between PA, PC. or PE (Figure 2D). Therefore, more PC is likely present on the tubules.

### 3.3. Longer Fatty Acid Chains Support Tubule Formation

Although the headgroup of a lipid likely plays a key role in determining localization to curved membranes [5,6,7,8,9], acyl chains have also been shown to affect the overall geometry and sorting on different membrane shapes [6,10]. To determine the role that acyl chains play in lipid sorting, several PAs with modified tails were compared according to their ability to sort into curved membranes and support the formation of tubules. Specifically, two fully saturated NBD-PAs (16:0 12:0-NBD PA and 16:0 6:0-NBD PA) were used in addition to the monounsaturated 18:1 12:0-NBD PA (Figure 3). The presence of 18:1 12:0- NBD PA led to more tubule formation than the presence of 16:0 12:0-NBD PA, and 16:0 12:0-NBD PA led to more tubule formation than 16:0 6:0-NBD PA, showing a trend in the size of the lipid tails (Figure 3A,B). PA supports the formation of more tubules when more carbons are present in the fatty acid chains, as shown by the higher density of tubules with PA compared to PC with identical acyl chains (Figure 3B). Unlike the density of the tubules, the intensity of the tubules (Figure 2C) did not trend with the number of carbons in the fatty acyl chains. Instead, tubules with 16:0 12:0-NBD PA appeared brighter than those with either 16:0 6:0-NBD PA or 18:1 12:0-NBD PA (Figure 3C); however, the variation observed in tubule intensities was large for all tubules. The size of the tubules formed did not vary as a function of the fatty acid tail, with no significant difference in the FWHM of the tubules noted between PAs (Figure 3D). Overall, this suggests that longer acyl chains and the incorporation of a bent, unsaturated tail on PA lipids can enhance the formation of membrane curvature, but the acyl tails do not affect the overall size distribution of the tubules that form.

### 3.4. Dithionite Quenching of NBD Reveals PA Localization to Negative Curvature

The presence of PA enhances tubule formation, but it is not clear from the imaging of STuBs which leaflet PA sorts into. To determine if PA prefers the inner leaflet (negative curvature) or the outer leaflet (positive curvature), a dithionite-quenching assay was performed with liposomes extruded through a 100 nm filter. Typically, the extrusion process yields a distribution of liposome sizes ranging from approximately 50–150 nm [10]. Dithionite quenches NBD fluorescence [31,32,33]; however, dithionite does not usually penetrate through a synthetic lipid barrier. Therefore, the outer leaflet is quenched preferentially. Figure 4 shows that dithionite quenched DPPE-NBD, a positive-curvature sorting lipid, to a greater extent that NBD-PC. This suggests a greater localization to the outer, positively curved leaflet. Conversely, all NBD-PAs tested were quenched to a lesser extent, supporting a localization to the negatively curved leaflet (Figure 4A–D). Melittin was used to form pores in the vesicles to quench the remaining inner leaflet NBD molecules (Appendix A) [6,34]. The fatty acid tails made no difference in the quenching assay (Figure 4D), suggesting that all PAs tested were similarly sorted to the interior of the liposomes.

## 4. Discussion

In this work, a new membrane curvature assay that leads to the formation of tubules protruding from a supported lipid bilayer [19] has been used to determine if PA prefers and/or stabilizes curved membranes. Although STuBs are simple to form and yield the visualization of lipids at regions with and without curvature simultaneously, there are challenges. To form STuBs, much higher concentrations of NaCl than is physiologically relevant are needed. Additionally, the read-out of the assay is fluorescence, necessitating fluorophores on the lipids. Studies using NMR experiments [32] and MD simulations [33] suggest that acyl NBD labels can bend towards the water interface. This likely complicates NBD-quenching assays (Figure 4) and could affect localization. To account for dye-labeling effects, the use of an identically labeled PC as a control was essential for comparison. Despite these challenges, the use of fluorescent lipids also provides advantages. Fluorescence assays have a high signal-to-noise ratio, allowing lower concentrations of lipids to be assessed, down to single-molecule-level concentrations [16]. In this work, the sensitivity of the assay allows PA curvature localization to be probed at a concentration that is similar to the amount of PA in mammalian and plant cells [20,21].

The formation of STuBs is straightforward and the addition of 500–1000 mM NaCl reliably forms tubular structures (Figure 1A,B), similar to what others have observed [19]. These structures have varying diameters and intensities (Figure 1C–E), which provides a method to observe the localization of dye-labeled lipids to varying curvatures. The wide distribution of tubule sizes (Figure 1G) provides an advantage over previous work using a nanoparticle-templated supported lipid bilayer [16], which contains only one size per sample as determined by the template choice. In the STuB assay, flat regions are also present and in continuum with curved membranes (Appendix A), which allows for a direct comparison to regions with curvature, overcoming a limitation of liposome-based curvature-sensing methods [10]. By having flat regions present, slight variations in the fluorescent lipid content when preparing STuBs or in microscopy, such as laser power, are internally corrected.

The STuBs assay was used to determine the sorting of phospholipids at curvatures relative to flat regions and whether certain lipids could aid in curvature formation. From the TIRF microscopy images, several features of the samples were quantified to determine if tails labeled PA and PC and headgroups labeled PE affected tubule formation or were recruited to tubules. First, the density of the tubules was determined to depend upon the lipid composition; lipids that support curvature (NBD-PAs and PE-NBD) led to the formation of more tubules when compared to the control, NBD-PC (Figure 2B). Second, the accumulation of fluorescently labeled lipids at tubule sites relative to the surrounding flat regions (ΔF/S) was measured. In this measurement, NBD-PA and PE-NBD (Figure 2C) accumulated slightly less than or at a similar rate to PC, and this depended on the tails on PA (Figure 3C). A direct comparison of the role of the headgroup was determined by comparing PC to PA, both with 18:1 12:0 NBD-labeled tails (Figure 2). PA stabilized curvature was observed in the increase in the number of tubules present (Figure 2B). However, the intensity of PA at tubule positions was slightly, but not significantly, less than PC (Figure 2C). Meanwhile, PE-NBD both stabilized tubule formation (Figure 2B) and was significantly less intense at tubule positions (Figure 2C).

We hypothesized that a reduction in the intensity of PE and PA relative to PC (Figure 2C) could be due to preferential localization to one leaflet, thus excluding the curvature-sensing lipids from a portion of the tubule bilayer and reducing fluorescence. To test this, a fluorescence-quenching assay was performed in liposomes extruded through 100 nm filters as shown in Figure 4. NBD-PAs quenched the least, followed by NBD-PC, then PE-NBD. This suggests that PA is protected from dithionite, which cannot penetrate the membrane to reach the inner leaflet; meanwhile, PC is quenched more, and PE-NBD is quenched the most (about 85%) and likely preferentially sorted to the outer leaflet (Figure 4D). Overall, we conclude that both NBD-PA and PE-NBD assist in the formation or stabilization of membrane curvature with NBD-PA sorting to the inner, negatively curved leaflet and PE-NBD sorted to the outer, positively curved leaflet. The sorting of the headgroup labeled PE to the inner, positively curved leaflet agrees with past work [5,6,9,30,35,36,37,38,39], although the preference for the curvature of PE-NBD may depend on the acyl chains of other lipids present [40].

Curvature-based lipid sorting is often discussed in reference to the lipid headgroup, with smaller headgroups preferring negatively curved lipid membranes and larger headgroups preferring positively curved membranes. However, the fatty acyl chains present on a lipid are essential for sorting within cells [41] and on curved synthetic membranes [10], with lysolipids showing a strong preference for positive curvature [5]. Conversely, previous studies have also demonstrated that lipids with more carbons in the acyl chains have a greater preference for curvature, where two-tailed lipids and longer lipid tails accumulate more at positively curved membranes [6,10]. In our past work, a lipid with two acyl chains (Fluorescein-DPPE) accumulated more at curvature than a single-tailed, fluorescein-labeled fatty acid (hexadecanoic acid) [6]. This suggests a mechanism that is different from the geometry of the lipid, and a “defect” site mechanism has been proposed [10,11,12]. In this model, the bent area in the positively curved membrane leads to the formation of packing defect sites, portrayed as a larger gap in the headgroups. This space can be filled with lipids or acylated proteins, where more carbon in the tails leads to more accumulation [10]. To test whether the tail composition affected PA accumulation, acyl-labeled PAs with varying tails were used, namely 16:0 12:0-NBD PA, 16:0 6:0-NBD PA, and 18:1 12:0-NBD PA (Figure 3A). PAs with shorter acyl chains formed significantly fewer tubules, with 16:0 6:0-NBD PA forming the fewest and 18:1 12:0-NBD PA forming the most (Figure 3B). The average size of the tubules did not depend on the tails, as all lipids yielded tubules that were approximately the same diameter (Figure 3D). However, the amount of NBD-labeled lipids that accumulated at tubule positions did not trend with the acyl chain length (Figure 3C). The saturated 16:0 12:0-NBD PA accumulated more than the unsaturated 18:1 12:0-NBD PA. This could be due to a more limited access to the positively curved leaflet, which is not supported by our data pertaining to NBD quenching within liposomes (Figure 4D). Instead, it is useful to note that the longest lipid is also unsaturated and, thus, bent, whereas 16:0 12:0-NBD PA and 16:0 6:0-NBD PA are both saturated lipids. Therefore, the interpretation of the accumulation (Figure 3C) could also be due to differences in tail saturation, with unsaturated lipids accumulating less at tubule sites. However, more lipids should be examined in future work to develop a model based on lipid unsaturation. Overall, the longer-tailed PA lipids and curvature-sensing lipids (PA and PE) both support the formation of more tubules.

In a complementary but independent assay, dithionite was used to quench NBD-labeled lipids to determine which leaflet of a membrane lipids prefer [31,32,33]. Using LUVs extruded through 100 nm pores, dithionite quenched more than 50% of all lipids tested. As a control that should not prefer curvature, NBD-PC fluorescence was measured (Figure 4D). NBD-PC was quenched more than expected by dithionite. This could be due to accessibility of the dye [32,33], and this is in line with previous studies that show the slow transport of dithionite across some membranes [31,32], but disagrees with others [40]. A second reason PC is quenched by more than 50% could be due to the liposome size; on small liposomes, the surface area on the outer leaflet is greater than the inner leaflet. However, we calculated this difference (Figure 4D, dashed line) and it does not account for the observed loss in fluorescence from dithionite treatment. Therefore, NBD-PC was used as a negative control to compare other lipids to because it is a lipid expected to have limited curvature preference [42]. PE-NBD was a positive control because several studies demonstrated that dye-labeled PE lipids have shown a preference for positive curvature in SLB studies and tubules extending from giant unilamellar vesicles [6,30]. However, another study in highly curved, small unilamellar vesicles (*d* < 100nm), only weak sorting was observed for PE-NBD, suggesting that the intrinsic shape of a lipid is not the only driving force for membrane curvature sorting [18]. The positive curvature preference we measure could be likely due to the dye on the headgroup altering its geometry or another mechanism, such as the defect site model [10]. When compared in the quenching assay, PE-NBD was quenched significantly more than NBD-PC (Figure 4D). This suggests that PE-NBD accumulated more on the outer leaflet of liposomes and was more accessible to dithionite treatment. Additionally, when compared to the NBD-PC control, PE-NBD formed more tubules (Figure 2B), suggesting a preference for curvature and possibly stabilization thereof. Unexpectedly, the PE-NBD tubules were significantly dimmer than the NBD-PC tubules (Figure 2C). One possible explanation is that PE-NBD localizes to the positive curvature specifically, while PC may be on both leaflets, and the liposome-quenching assay supports this hypothesis. Meanwhile, dithionite quenched fluorescent PAs to a lesser extent in the liposome assay (Figure 4), and 18:1 12:0-NBD PA tubules were dimmer than 18:1 12:0 NBD-PC tubules, although not significantly (Figure 2C). Following the same reasoning as above for PE, we conclude that PA is likely sorted to the inner leaflet of liposomes. In fact, all three PAs with varying tails quenched to the same value in the presence of dithionite, suggesting a similar preference for negative curvature (Figure 4). Together, these data suggest PE-NBD predominantly localizes to the outer, positively curved leaflet of STuBs and liposomes, whereas PA-labeled lipids prefer the inner, negatively curved leaflet, in agreement with others [6,30,31,39]. Overall, STuBs are a new method for measuring the curvature sorting of lipids and curvature stabilization, and PA and headgroups labeled PE are curvature-stabilizing lipids.

## Figures and Tables

**Figure 1 biomolecules-12-01707-f001:**
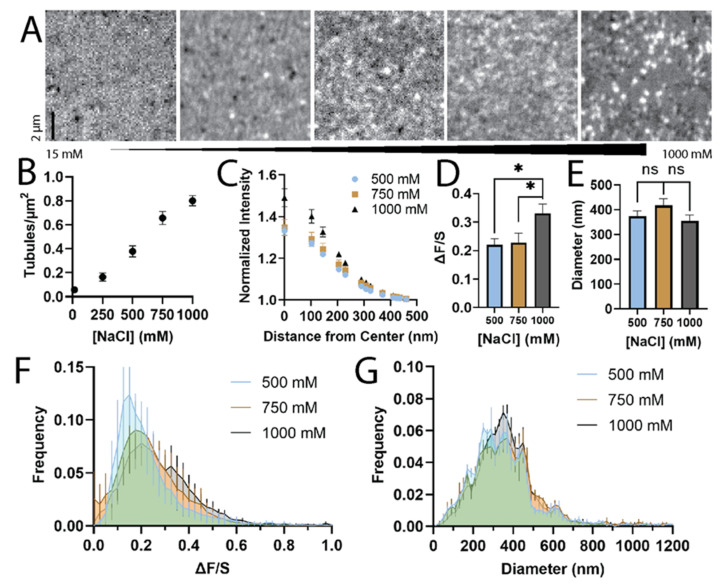
Tubules form in the presence of 500–1000 mM NaCl. Lipid tubules were characterized using 0.1% DiD as a lipid marker. Increasing concentration of NaCl increases number of tubules per area. (**A**) Tubules were imaged using confocal microscopy. [NaCl] from top to bottom (mM): 15, 250, 500, 750, 1000. All images are autoscaled. (**B**) DiD-labeled tubule density as a function of NaCl concentration. (**C**) Radial plots of intensity of DiD at 500 mM (blue circles), 750 mM (brown squares), and 1000 mM (black triangles), normalized to intensity at 951 nm from center. (**D**) Average intensity of lipids at sites of tubules for different [NaCl], displayed as ΔF/S, a function of tubule intensity and surrounding intensity. Significant differences noted by *, where a *t*-test *p* value < 0.05. (**E**) Average size in nm at [NaCl] of 500, 750, and 1000 mM. All error bars are SEM (*n* = 9). No significant (ns) differences measured. (**F**) The distribution of intensities of single tubules. (**G**) The distribution of the diameter, as measured from the FWHM of the imaged tubule. Error bars in histograms are SEM from three days.

**Figure 2 biomolecules-12-01707-f002:**
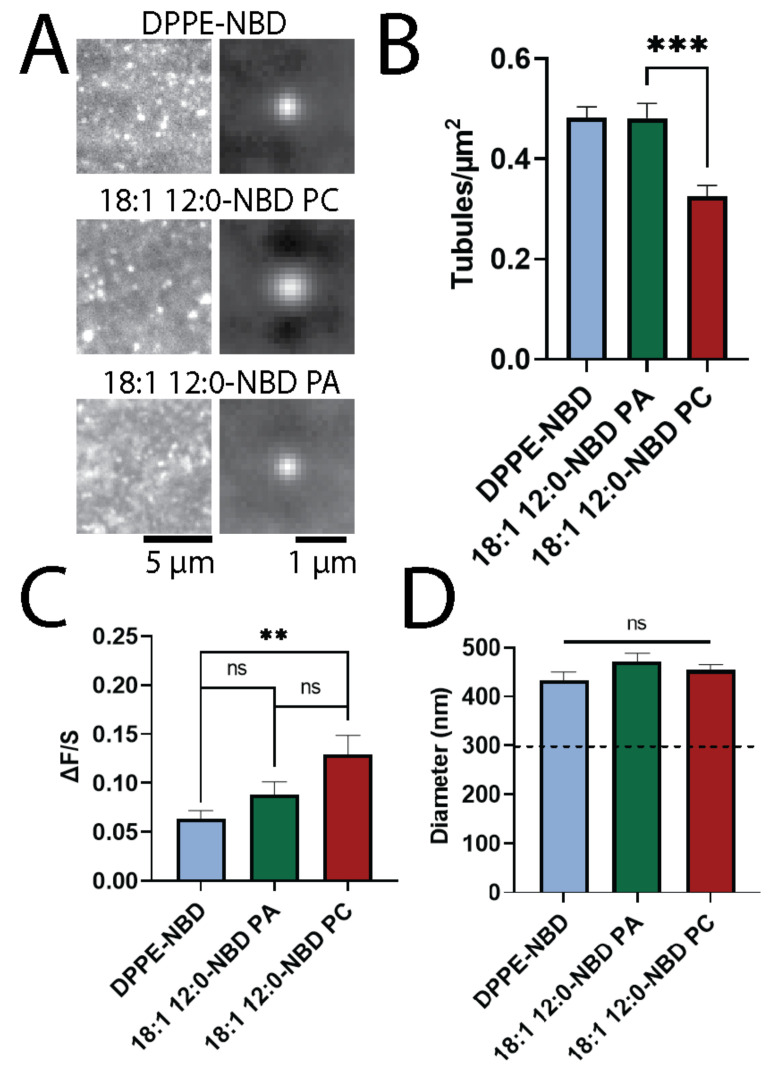
Tubulated lipid bilayers with PA and DPPE induce more tubule formation than PC. (**A**) Examples of bilayers with 18:1 12:0-NBD PA (top), 18:1 12:0-NBD PC (middle), and DPPE-NBD (bottom), with examples of STuBs (left) and averaged tubules (right). (**B**) Average tubule density of bilayers with respective lipids. (**C**) Average ΔF/S of lipids. (**D**) Average diameter (FWHM) of lipids. On our microscope, the average FWHM of diffraction-limited 200 nm green polystyrene nanoparticles is 296 nm (dashed line). Error bars are SEM. *n* = 12 membranes between 3 days. ** *p* < 0.005, *** *p* < 0.0005 and ns = not significant on a *t*-test.

**Figure 3 biomolecules-12-01707-f003:**
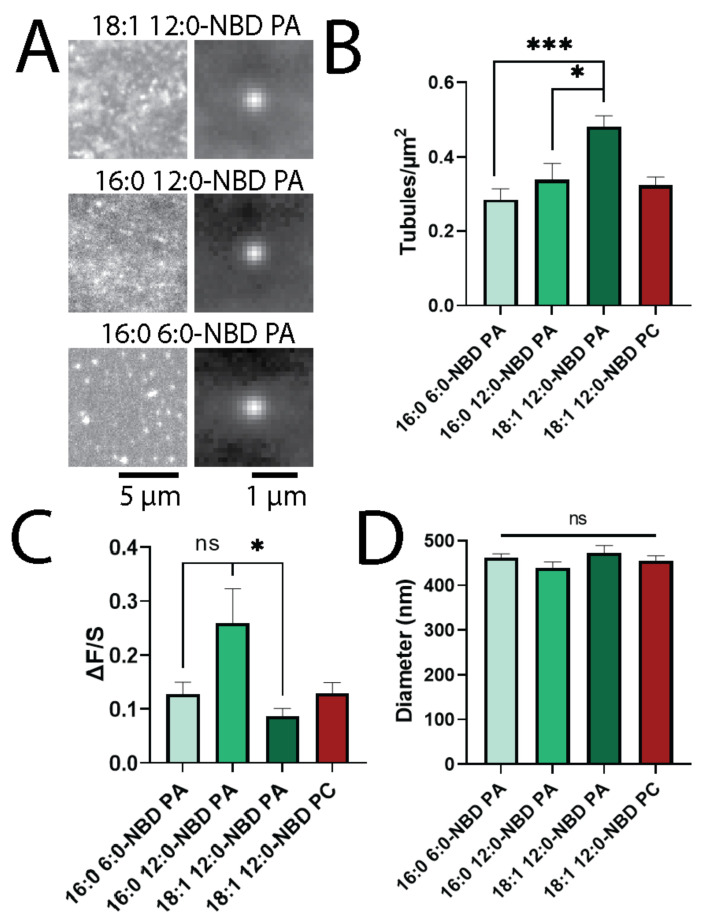
Lipids with longer fatty acid chains form more tubules. (**A**) Examples of tubules containing 1% fatty-acid-labeled PA. 18:1 12:0-NBD PA (top), 16:0 12:0-NBD PA (middle), 16:0 6:0-NBD PA (bottom). (**B**) Tubule density of bilayers with respective lipids. (**C**) Average ΔF/S values of lipids. (**D**) Average FWHM of lipids. All differences are not significant in D. Error bars are SEM. *n* = 12 membranes between 3 days. * *p* < 0.05, *** *p* < 0.0005 and ns = not significant on a *t*-test.

**Figure 4 biomolecules-12-01707-f004:**
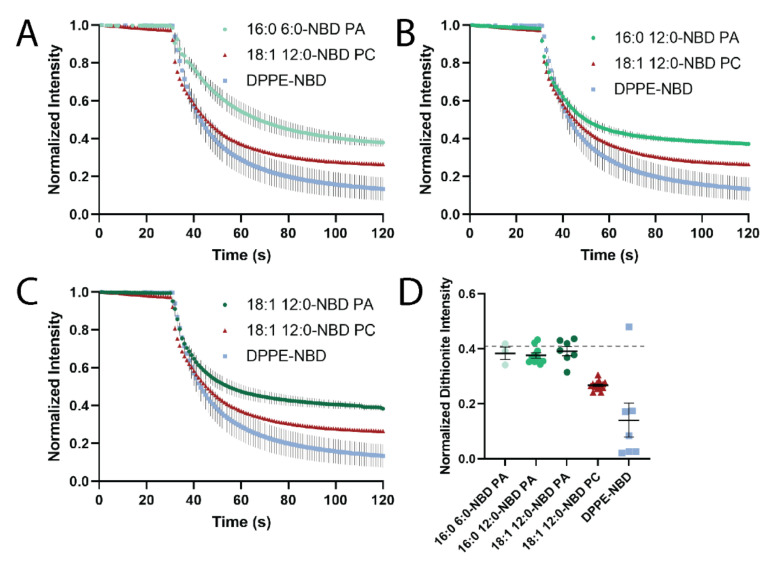
PA localizes to negative curvature in a liposome-based dithionite assay. Liposomes containing 98% POPC, 1% DOPE-PEG, and 1% NBD-labeled lipids were extruded through 100 nm filters and fluorescence was measured in a fluorimeter. After 30 s, dithionite was added. (**A**–**C**): the fluorescence traces over time. (**D**): Normalized intensity after addition of dithionite. Dashed line: theoretical intensity if only the outer leaflets were quenched and liposomes were symmetric, accounting for surface area and dilution. Error bars are SEM; error on PC was smaller than the data points.

## Data Availability

Not applicable.

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
