# Peer review of "Phosphatidic Acid Accumulates at Areas of Curvature in Tubulated Lipid Bilayers and Liposomes"

_biomolecules, 2022, doi:10.3390/biom12111707_

Round 1

Reviewer 1 Report

The article entitled “Phosphatidic acid accumulates at areas of curvature in tubulated lipid bilayers and liposomes” proposed by Bills and Knowles highlighted and studied how one lipid, phosphatidic acid, could accumulate in curvature areas. For this purpose, they used StuBs created by different deposition of liposomes on surface with varying concentrations of NaCl. The article is of interest to study rules of the different phospholipids in areas of curvature involved in many important cell functions. Nevertheless, the article lacks comparison with actual methods used to study such phenomena. The images shown have to be improved in term of size and quality. The authors should explain the inputs of this new technique as compared to the results they published in 2017 and those from Kamal et al published in 2009 with SUVs. The authors should complete the results obtained through other complementary techniques such as TEM.

Abstract :

L3 : and has a net negative charge

L4-L8 : the authors should separate this long sentence into 2 or 3 shorter ones to simplify the text.

Introduction :

L4 : Arabidopsis thaliana

3e paragraph L5 : why the term “higher” is in italics?

3e paragraph L8-L10 : this sentence should be rewritten and/or separated into two shorter sentences since it contains 2 different ideas.

4e paragraph L1: the word protein lacks a s

4e paragraph L6 : demonstrated. The authors have here to insert a reference or explain if the sentence is linked to the present study.

5e paragraph L4 : microscopies instead of microscopy

Methods

2.1 L3 : mL; all the lipids should be written the same way, including the reference for each;  depending on techniques, the liposomes and SLBs contain different percentage in POPC ? why ? how the authors measured liposome diameter ?

2.3 if the home-built alignment code in Matlab has already been used in previous studies, the authors should add these references. What is the signification of the FWHM ?  The authors have to explain why the larger tubules were not measured in this study and others.

Results

3.1 1st paragraph L9 : recovered

The first paragraph lacks analysis of the results obtained and shown on Figure 1F and 1G. Moreover, since these 2 figures have been generated from 9 different experiments, they should contain also SEM and error bars.

The authors have to explain why they started the study using a very low concentration in NaCl.

3.2 2nd paragraph : the sentence “This suggests that ….” has to be rewritten and reformulated. The figure 2A was not analyzed and in particular, the difference between the 3 different conditions. Legend of Figure 2: the authors gave here the size of diffraction-limited green polystyrene nanoparticles. They should include these nanoparticles within the methods part and give the real size of those commercial nanoparticles.

3.3 1st paragraph : to determine the role of acyl chains in lipid sorting. The authors have to explain why they used saturated lipids for this part of the study. The 2 sentences starting by the word “overall” have to be rewritten.

3.4 1st paragraph : one space should be added before [27-29]. The error bars lack on the figures 4A, B and C. Some authors used detergent to dissolve the membrane and drop the fluorescence to zero. The authors should explain why they only used dithionite. The authors have to explain/compare the behavior obtained for NBD-PC.

Discussion

L4 : figure is not written the good way. similar to others already observed

1st paragraph: the authors should rewrite the last sentence

2nd paragraph : the authors gave here the size of liposomes but never gave before in the methods part this information nor the technique to determine it. The very long last sentence has to be rewritten and shorten.

3rd paragraph : the sentence explaining the link between size of tubules (Fig. 3D) and the amount of NBD labeled lipids (Fig3C) is contradictory and has to be rewritten. The last sentence should be rewritten since both elements (longer tailed PA lipids and curvature sensing lipids) support the formation of more tubules.

4th paragraph : one or more words are missing after quench in the first sentence. This last paragraph should be rewritten to focus on the results obtained for NBD-PA in regard to those obtained with both controls NBD-PE and NBD-PC.

The conclusion of the article should be extended and include advantages/drawbacks of such StuBs for the  measurements /studies of curvature sorting of lipids. It has to be in conformity with the conclusion of the introduction.  

Author Response

Please see the attached file for point by point responses. All suggestions led to changes in the manuscript. 

Reviewer 2 Report

In this manuscript, the authors investigated the relationships between membrane curvature and PA localization by total internal reflection fluorescence microscopy and by the dye quenching assay using liposomes.  However, the methods using fluorescently labeled lipids have important limitations.

[1] In this study, the fluorescently labeled PA is used to investigate the localization of PA in the membranes.  However, the fluorescent probe NBD affects the molecular structure and physicochemical properties including hydrophobicity.  The NBD moiety is hydrophilic and may loop back to the head group region in the membrane bilayer.

See the web site of the products (https://www.thermofisher.com/jp/ja/home/references/molecular-probes-the-handbook/probes-for-lipids-and-membranes/fatty-acid-analogs-and-phospholipids.html#head1).

Thus, the molecular shape of NBD-labeled PA in the membrane may be largely different from that of unlabeled PA.  The authors should discuss the limitations of the assays using fluorescently labeled lipids.

[2] In ABSTRACT, the authors have described “Using small unilamellar liposomes in a dye quenching assay”.  However, in general, liposomes prepared by the extrusion through 100 nm filter are called large unilamellar liposomes.  Small unilamellar liposomes (~30 nm in diameter) are prepared by sonication.

[3] In total internal reflection fluorescence microscopy, ΔF = circle – annulus.  What are “circle” and “annulus”?

[4] In Figs. 1G, 2D and 3D, what is FWHM?  Whose sizes were measured?

[5] In Supplementary Materials (page 9), Table S1 and Video S1 are not attached.

Author Response

(The authors gave the same response as above.)

Round 2

Reviewer 1 Report

Article MDPI

The authors submitted a revised version of their article entitled “Phosphatidic acid accumulates at areas of curvature in tubulated lipid bilayers and liposomes”. The images have not been improved in term of size and quality. They included comparisons to Kamal but not to their results published in 2017. Moreover, no complementary experiments have been performed. 

Abstract :

All changes performed

Introduction :

Most changes have been performed except inside the 4th paragraph; the authors have to reformulate the sentence concerning the study described. Is this study the present one or the one described in the reference 19 ?

Methods

Half of the changes have been performed. The authors have to reformulate the explanation they gave for the difference of percentage and to give more details on low intensity and compensation. Moreover, the authors did not explain how they measured liposome diameter. In addition, authors added an additional sentence on MB-DPPE; this sentence and the one before have to be changed if both DiD and MB-DPPE are used for fluidity. They have to explain also why they use two different compounds and what are the difference between them. For larger tubules, the authors should include in the methods part the explanation given in the result part.

Results

Half of the changes have been performed. Concerning the figures 1F and 1G, the authors have deleted them from the manuscript to place them as supplementary figures. Why ? Since the information given by these 2 parts of the figure concern results shown on Figure 1. The authors have also to reformulate the last sentence of the first paragraph concerning these figures. Indeed, the difference between the 3 concentrations is not so important since there is overlap between error bars (fig s3 A and C).

In the second paragraph, the sentence “in this experiment”, replace are by were for the lipids. The figure 2A was not analyzed and in particular, the difference between the 3 different conditions.

In the third paragraph : the authors have to reformulate the first sentence since it highlights previous studies showing the possible role of the head group of lipids. Moreover, this sentence should be written in a past time. One symbol of no significant difference (ns) should be added on figure 3D.

In the fourth paragraph, the authors did not give information about the size of the liposomes and did not explain/compare the behavior obtained for NBD-PC. The title of figure 4 has to be conjugated at past time. Moreover, they added experiments with melittin. The additional experiments have not been explained in the method part; moreover, they did not explain the nature and function of this compound; they have to add references to support the use of such compound instead of detergents most often used in such studies. Finally, the data obtained with this compound should be added to the corresponding conditions in Figure 4 (with error bars) to allow comparisons between the studies in presence and in absence of melittin.  

Discussion

Less than half of modifications have been performed.

The authors added a paragraph on StuBs, on their advantages and drawbacks. This paragraph should be placed just after the first sentence of the discussion since it introduces on the supported lipid bilayer used.

The authors change the sentence about the size of liposomes and only explained that they were extruded through a 100 nm filter. They should perform experiments and determine this diameter, show if the population of liposomes generated is homogenous or heterogenous.   

In the last paragraph : in the sentence starting with “PE-NBD was a positive”, the verb should be in a past time : demonstrated. It’s also the case for the verb measure in the next sentence. The authors have to harmonize the ways they wrote NBD-PE and NBD-PC. In the sentence starting by “meanwhile” the term “its” placed before tubules has to be changed to a proper one.

Author Response

Please see the attached file for a full response.

Reviewer 2 Report

In Supplementary Materials, there are two Supplemental Figure 1.  Either figure number should be corrected.

Author Response

Thank you for the constructive comments. The comment is in italics and the response is not. The text changes will be cut and pasted here and shown in a separate document in the manuscript with track changes.

In Supplementary Materials, there are two Supplemental Figure 1.  Either figure number should be corrected.

This has been corrected.